# Hedonic pricing analysis for semen of dairy bulls in Brazil

**Glauco Rodrigues Carvalho**[1], **Weslem Rodrigues Faria** [2]*, **Vinícius Pimenta Delgado Ribeiro Nardy** [2], **Admir Antonio Betarelli Junior**[2]

1 Empresa Brasileira de Pesquisa Agropecuária (EMBRAPA), Juiz de Fora, MG, Brazil, 2 Faculdade de Economia, Campus Universitário, Universidade Federal de Juiz de Fora, Juiz de Fora, MG, Brazil

* weslem.faria@ufjf.edu.br

**Data Availability Statement:** All relevant data are within the paper and its Supporting Information files.

**Funding:** Grants: 1) Conselho Nacional de Desenvolvimento Científico e Tecnológico (CNPq).

## Abstract

Dairy farming is an important aspect of the Brazilian agricultural sector. The presence of numerous producers has created a large number of jobs in this field, contributing significantly to the rural economy. Artificial insemination (AI), used as one of the main means of reproduction, is increasingly gaining importance in the genetic improvement of animals. Given this scenario, the semen of bulls has become extremely marketable and an important aspect of the animal industry. This study aims to develop a hedonic model for the price of semen doses of dairy bulls based on the information from the main sellers of the product in the Brazilian market. The main findings reveal that there is an additional premium for proven bulls. Semen doses from Gir bulls proved to be more expensive, and the AI firm ALTA has a discount compared to the other firms. From the characteristics obtained in the tests, there is a premium only for the Predicted Transmitting Ability–Milk (volume). The most valued feature found is the dose being sexed, that is, the dose that guarantees the gestation of a female from its use. Semen doses from dead bulls has proved to be more expensive.

## 1 Introduction

In 2017, the agricultural sector was responsible for 64% of the positive variation in Brazil's gross domestic product (GDP) [1]. Part of this advancement was achieved through the impulse of the national bovine industry. According to the Municipal Livestock Survey (PPM, IBGE), there were 214.9 million head of cattle in the country in 2017, 12.6 million more than in 2008, [2]. In Brazil, 12.13 million doses of bovine semen were sold in 2017. This number is 62% higher than the number observed in 2008 (7.46 million doses). In 2017, semen production was 8.15 million doses and imports were 5.88 million doses [3].

The big boost in the cattle industry started with the rise of reproductive biotechnologies. Artificial insemination (AI) plays an important role in animal breeding because it is the main means of disseminating genes in the world [4]. AI is the method of choice for countries with the highest milk production to insert genes into the population, as it is a simple, economical and easy-to-disseminate method compared to other methods such as embryo technologies and natural breeding [5]. Moreover, AI includes both quantitative and qualitative increase in production through the genetic improvement of the herd. The impact of using AI can be

Award number: 302792/2019-8. Recipient: Weslem Rodrigues Faria. 2) EMBRAPA Gado de Leite – Juiz de Fora. There is no Award number. Recipient: Glauco Rodrigues Carvalho. 3) Universidade Federal de Juiz de For a (UFJF). There is no Award number. Recipient: Weslem Rodrigues Faria. The funders had no role in study design, data collection and analysis, decision to publish, or preparation of the manuscript.

**Competing interests:** NO authors have competing interests

measured by analyzing the results obtained in the USA. Over a period of 20 years (1955–1975), individual production increased from 2,415 to 4,706 liters of milk per cow. More recently, the average production has exceeded 8,500 liters of milk per cow [6].

Sexed semen is one of the main points within the discussion of artificial insemination. Most of the published works on semen from dairy cattle basically focus on two aspects related to economic yield and milk production. The first concerns the yields of different breeds, considering genetic manipulations, crossbreeding and experiments on the evolution of specific breeds from sexed semen. Evidence found indicated that the combined use of sexed semen for purebred and conventional bovine semen for terminal crosses improves meat production in dairy herds [7]. The genetic crossing of dairy cattle breeds is an increasingly evident practice, but the consequences, mainly genetic, are little known. [8] considering different breeding scenarios, breeds, possibility of genomic testing and sexed semen indicated that the breeding could result in lower genetic returns in the whole herd compared to pure breeding scenarios, but that it would be possible to obtain additional economic return from the combined practices.

The second aspect concerns the economic evaluation of the use of sexed semen in artificial insemination. One of the main issues related to the use of sexed semen concerns the pregnancy rate in relation to the use of conventional semen. According [9], pregnancy rates in cattle with 2 million sexed sperm per insemination dose are about 80% of those with conventional semen at normal sperm doses. Even so, about 2 million doses of bovine semen applied annually in the United States are sexed, since the expansion of dairy heifers is crucial for the dairy sector and improvements made in sexing procedures have increased fertility. Overall, evidence indicates that artificial insemination with sexed semen tends to increase the economic profit of dairy farmers compared to artificial insemination with conventional semen [10, 11].

The cost-benefit analysis by [12] indicated that sexed semen can facilitate faster and more profitable expansion of the dairy herd, increasing the number of dairy heifer births, but market prices and the type of management must be considered when deciding on its use. The study by [13] indicated an increase in the genetic level and in the net economic return with the use of sexed semen from different ones in 5 herds at different levels of management. [14] found evidence that the use of sexed semen in artificial insemination produces positive economic effects by increasing the herd of females. [15] indicated that the use of sexed semen brings financial advantages to a system based on pastures, but that these advantages are sensitive to the herd fecundity base, being smaller as the fecundity is lower.

In this context, it is necessary to consider the producers' preference regarding the characteristics of the bulls chosen for the breeding process. [16] indicated that factors such as technology, which includes information about semen sexing, and conception rate have positive effects on the dairy industry. Historically, dairy production around the world has been focused on increasing the milk production (such as performance indicators for cow production), followed by the gradual inclusion of characteristics of the milk component, such as protein and fat content [17]. [18] emphasized that a predominantly greater focus is given to the volume produced. More recently, [19] also found evidence that the volume produced is the component most used in the selection of milk herds. However, this work showed the importance of including factors such as protein and fat in animal breeding programs, because they can increase the genetic and economic efficiency of Brazilian herds.

In a study for Brazil, [20] indicated that the Holstein strains considered had the highest PTAs (Predicted Transmitting Ability) for milk production, but the lowest PTA for the percentage of solids in milk. [21], also under study for Brazil, indicated that the average milk production of the daughters of the bull with the lowest PTA was 20% less than that of the daughters of the bull with the highest PTA. According to [22] the average rate of genetic gains

in terms of PTA has been growing annually since 2010 when genomic tests became more wide-spread to select bulls, reaching in the period 2013–2017 a higher value in more than twice the value of the period 2003–2007.

According to [23], in the five largest Brazilian insemination centers that commercialized bovine semen, only one third of the bulls offered were included in public or private genetic evaluations or presented in the promotional material information about their performance in relation to their contemporaries. [23] indicated that the information on the genetic value for milk production and its components had little influence on prices, which were more related to the kinship of famous bulls. However, [21] found evidence that the use of semen from proven bulls resulted in higher milk production compared to the national average result.

With a change in the scenario in which breeding technologies are being increasingly used, the evolution and application of progeny tests have increased the genetic information of each bull. It is expected to obtain a greater clarification on how semen prices of dairy cattle are formed. Thus, this study aims to infer the relationship between the characteristics of the bull and the price of the semen using a hedonic pricing model. One of the main objectives is to analyze and quantify how the attributes affect the price of semen from milk bulls in addition to identifying whether proven bulls have higher prices than the others. It is intended to generate this information for the Gir, Girolando, and Guzerá breeds. No studies were found in the literature that had these same objectives. The most similar work is that of [24] that was carried out for the Holstein breed in Alberta, Canada.

Hedonic prices are defined as the implicit prices of attributes and are revealed to economic agents based on the prices attributed to differentiated products and the specific quantities of characteristics associated with them [25]. A similar analysis strategy was recently used by [26] for an analysis of the Italian yogurt market, [27] for the meat restaurant market in Seoul, South Korea, [28] for the milk market in Hawaii and [29] for the ultra-high-temperature milk market in Italy.

More recently, some studies have also used hedonic pricing models to assess the milk and dairy market and consumer preferences. [30] evaluated implicit prices from different origins for fluid milk in Italy. Similarly, [31] analyzed the marginal prices associated with different countries of origin in the Chinese dairy market. [32] evaluated the effects on prices of the declaration of origin of food products considering a group of European countries. Overall, these studies found evidence that the declaration of origin would have positive effects on milk or food prices. The consumer's perception of the product's quality and the "healthy" characteristic are attributes increasingly demanded by consumers. In this sense, [33] indicated that consumers tend to pay premiums for nutritional attributes added to milk such as vitamins. [34] analyzed the Chinese market and the results found by this study showed that attributes of perception of nutritional content and green food have positive effects on food prices. Finally, positive effects on milk prices were found for attributes that contained information on animal welfare [35].

This study is divided into three sections, in addition to this introduction. The next section describes the data and model. The results have been presented and discussed in the following section. Finally, the conclusions provide some suggestions for the dairy sector.

## 2 Materials and methods

### 2.1 Data

The data used have been collected directly from the heads of each semen selling center. There are prices of 554 semen from milk bulls (including sexed semen) of Gir, Girolando, and Guzerá breeds, whose statistical information takes into account the main AI firms in Brazil—

**Table 1. Statistics and description of the variables collected on bovine semen (price in R$ /dose of semen).**

| Variables | Obs | Mean price | SD | Minimum price | Maximum price | Variable description |
|---|---|---|---|---|---|---|
| Price | 554 | 57.10 | 113.87 | 14.00 | 2019.30 | Price of 1 semen dose |
| Breeds | | | | | | |
| Gir | 282 | 66.69 | 149.62 | 14.00 | 2019.30 | Gir = 1; others = 0 |
| Guzerá | 41 | 49.27 | 45.58 | 17.00 | 22.40 | Guzerá = 1; others = 0 |
| Girolando | 231 | 46.78 | 56.85 | 14.00 | 563.50 | Girolando = 1; others = 0 |
| AI firms | | | | | | |
| ALTA | 182 | 41.47 | 83.47 | 16.00 | 800.00 | ALTA = 1; others = 0 |
| CRV | 176 | 86.36 | 176.71 | 15.00 | 2019.30 | CRV = 1; others = 0 |
| ABS | 113 | 40.94 | 28.20 | 14.00 | 100.00 | ABS = 1; others = 0 |
| SEMEX | 83 | 51.31 | 39.45 | 22.00 | 250.00 | SEMEX = 1; others = 0 |
| Proven | | | | | | |
| Yes | 123 | 99.97 | 221.23 | 16.20 | 2019.30 | Yes = 1; others = 0 |
| No | 431 | 44.86 | 45.96 | 14.00 | 563.50 | No = 1; others = 0 |
| Sexed | | | | | | |
| Yes | 134 | 150.83 | 200.66 | 75.00 | 2019.30 | Yes = 1; others = 0 |
| No | 420 | 27.19 | 24.99 | 14.00 | 306.00 | No = 1; others = 0 |
| Alive | | | | | | |
| Yes | 223 | 51.25 | 63.61 | 15.00 | 700.00 | Yes = 1; others = 0 |
| No | 43 | 162.30 | 346.05 | 14.00 | 2019.30 | No = 1; others = 0 |

Source: Own elaboration based on research data. Obs—Observations; SD—standard deviation.

ALTA, CRV, ABS, and SEMEX—as shown in Table 1. The information is for June 2019. Moreover, information regarding the genetic attributes of the bulls already participating in this sample was extracted from the catalogs of Embrapa Gado de Leite's Progeny tests. This paper only uses information about market prices and characteristics of the bulls and the semen of the bulls. Therefore, for this paper no animal experiments were performed (e.g., anesthesia, euthanasia, and animal sacrifice).

The data is constructed based on two processes. The first refers to the genomics that represents the genetic mapping of the bull. Usually, the DNA of the bull is obtained from the animal's hair. In this process, the test results of the bulls are compared with the results of the benchmark bulls. Benchmark bulls are those that have been proven to generate offspring with high performance in terms of characteristics such as lactation, milk production, milk solids content, et cetera. Thus, this comparison provides evidence about the bull's ability to generate offspring that may or may not have high performance, based on the characteristics of the milk. The second process refers to the progeny test that represents the monitoring of cows in herds. In this test, cows and their offspring are monitored for lactation, milk production, milk solids content, teat size, hoof size, posture, and profile. Controls over the external environment, such as temperature, humidity, and feed, are used to isolate the genetic effect and characteristics of bulls (phenotype) on the cows' ability to generate high performance in terms of milk characteristics. The evaluation of the PTA (Predicted Transmitting Ability) is one of the main ways to verify the genetic transmission capacity of the bull for the high-performance generation in terms of the characteristics of the milk of its offspring. Thus, PTA represents the genetic value of the bull.

In addition to the price, the following characteristics have been analyzed: race (dummy), AI firm (dummy), number of herds, PTA—Milk, PTA—Age at First Birth, PTA—Fat (in kilos and percentage), PTA—Protein (in kilos and percentage), PTA—Solids (in kilos and

percentage), whether the semen is sexed or not (dummy), and if the bull is proven (dummy) and alive (dummy).

Table 1 presents the predominance of Gir bulls in relation to the other breeds. Together, the Gir and Girolando bulls represent more than 92% of the total number of bulls. There are only 41 Guzerá bulls. Among the AI firms, the most relevant are ALTA (32.9%) and CRV (31.8%).

Semen from proven bulls have an average price that is 122.85% higher than that of the unproven bulls. The semen with the characteristic "sexed", which guarantees the gestation of a female, has an average price that is higher than the non-sexed one: R\$ 150.83 per dose against R\$ 27.19 for the traditional one. The bulls that are still alive have an average semen price that is R\$ 111.05 per dose lower than those that are already dead.

## 2.2 Model

The method for assessing the value of product characteristics or components has changed since [36] work, in which the objective was to evaluate how the quality factors (e.g., color, size, and condition) of some products sold at fairs (e.g., tomatoes, asparagus, and cucumbers) were correlated with their prices. The analysis indicated high correlation coefficients (in module), demonstrating that these factors were associated with product prices. This seminal effort opened fronts for developments based on more sophisticated statistical techniques, with application to various themes, such as milk [37], cotton [38], fish [39], and wine [40]. Additionally, economic theory provided the basic foundation for guiding agents' behavior in market situations. Therefore, the analysis that is intended to be developed in this paper is methodologically justified by the analytical capacity of the hedonic price approach, with applications to evaluate various markets, and its development in the light of the economic theory that guides it. The main theoretical aspects are described below. It is interesting to note that [36] analysis was based on information collected via field research that focused on the commercialization segment. This research was conceived in the same context.

The theoretical background underlying the analysis is the economic theory of hedonic prices, comprising representative consumer decisions that consider the implicit characteristics of products in the utility maximization process. Any differentiated product can be considered a set of several quality attributes that consumers independently assess at the time of purchase. Thus, given a representative consumer's budget constraints, they consider the attributes of the product during the consumption decision-making process and obtain utility from that [41]. For instance, dairy producers, when purchasing bovine semen, expect to maximize utility by considering the product characteristics, such as tested semen and sexed semen. Furthermore, semen buyers may experience different satisfaction levels if they are aware of the different genetic transmission capabilities of the semen, such as the potential for higher milk volume production or higher protein content.

[41, 42] formalized a model based on consumer theory to evaluate the behavior and milk consumption decisions of a representative agent. The idea presented in these works is identical to that in this study, albeit the present research applies the idea to the bovine semen market. Works prior to [41, 42] formalized a model based on the theory of the firm, since analyses of the milk market focused on production techniques and the use of inputs to define the profit maximization behavior of a representative agent (e.g., [43–45]). Examples of this line of analysis are the employment of hedonic price models to assess the implicit effect of the use of each production input (e.g., [46]) or the use of milk as a production input to produce another product, such as cheese or yogurt (e.g., [37]).

Therefore, the hedonic pricing model aims to establish a relationship between the characteristics of a product and its price, that is, how much more or less the consumer (in this case,

milk producers) is willing to pay for a particular attribute. Although these prices are not explicitly expressed in the market, they can be estimated using a regression model capable of directly expressing the price of a product as a function of directly or indirectly observable quality characteristics [26]. At the end of the formalization presented below, the equations to be econometrically estimated will be derived.

Based on the formulations of [47] and [25] as well as the derivations of [41, 42, 46, 48], the hedonic price function considers a set of attributes that can be represented by a $Z$ vector of characteristics:

$$Z = (Z_1, Z_2, \ldots, Z_n) \tag{1}$$

Thus, the utility function is specified according to the characteristics of the product instead of the quantity of the product consumed. A representative agent's utility function, considering that $z_j$ is the attribute $j$ of the consumed product, is expressed as:

$$U = U(z_1, z_2, \ldots, z_n, \alpha) \tag{2}$$

where $\alpha$ is a fixed parameter of consumer preferences. It is assumed that the magnitude of a characteristic is a function of the product quantity and the number of product characteristics:

$$z_j = F(v_1, v_2, \ldots, v_i, z_{j1}, z_{j2}, \ldots, z_{jn}) \tag{3}$$

where $v_i$ is the quantity of product $i$ and $z_{ij}$ is the magnitude of characteristic $j$ in a unit of product $i$.

Therefore, the problem of maximizing consumer utility is represented by the Lagrangian maximization problem:

$$L = U(z_1, z_2, \ldots, z_n, \alpha) - \lambda\left(\sum_{i=1}^n p_i v_i - I\right) \tag{4}$$

where $p_i$ is the price of the product $i$, $I$ is the representative agent's income, and $\lambda$ is the Lagrange multiplier. The first order conditions in Eq (4) yield the following result:

$$\frac{\partial L}{\partial v_i} = \sum_j \frac{\partial U}{\partial z_j} \frac{\partial z_j}{\partial v_i} - \lambda p_i = 0 \tag{5}$$

Eq (5) provides the necessary elements for analyzing the value of the product characteristics. The term $\frac{1}{\lambda} \frac{\partial U}{\partial z_j}$ represents the implied price or marginal value of characteristic $j$. The term $\frac{\partial z_j}{\partial v_i}$ is the marginal value that characteristic $j$ gives a unit of product $i$. Isolating $p_i$ in Eq (5), we obtain the result [25] proposed, which specifies prices as a function obtained at the point of equilibrium between demand and supply of product $i$:

$$p_i = G(Z_j) \tag{6}$$

where $G(.)$ is a function of hedonic prices specified in general terms of the attributes associated with product $i$.

Eq (6) indicates that the price $p$ paid is a function of the marginal monetary value of the $j$-th attribute $Z$. The implied marginal price that a consumer is willing to pay for attribute $j$ is equal to the marginal cost the producer incurs to offer it [49].

[41] presented a concept capturing consumer perception of the value of the product as reflected in the price paid to acquire it. For example, if a consumer perceives $n$ characteristics of a product, then the amount paid for the product can be broken down as follows:

$$p = \sum_{j=1}^n V_j Q_j \tag{7}$$

**Table 2. Models and variables used in the estimations.**

| Models | Estimations | Dependent variable | Independent variables |
|---|---|---|---|
| Model 1 | Three estimations: 1) one for Gir, 2) one for Girolando, and 3) one considering Gir + Girolando | *Price* | *Proven* |
| Model 2 | One estimation | *Price* | *Breeds* (Girolando and Guzerá), *AI firms* (CRV, ABS, SEMEX) (Note: Gir from ALTA was considered the baseline product.) |
| Model 3 | Two estimations: 1) one for Gir and 2) one for Girolando | *Price* | *Number of Herds*, *PTA—Milk (volume)*, *PTA—Age at First Birth*, *PTA—Fat (kg)*, *PTA—Protein (kg)*, *Sexed*, *Alive* |

Source: Own elaboration based on research data.

where $V_j$ is the implicit value or price the consumer assigns to a unit of characteristic $j$, and $Q_j$ is the number of units of characteristic $j$.

An empirical representation of the decomposition of the product price is conceptualized in Eq (7), with a view to statistical estimation. Due to the data structure, three models are estimated. The main reason for estimating three different models is the limited availability of information about the progeny test for the Guzerá breed. The estimated models are described in Table 2.

The first model to be estimated aimed only to identify the implicit premium of the semen price with respect to the progeny test. Three estimations were performed considering the specification of the first model: one for the Gir breed, one for the Girolando breed, and one considering the bulls of both breeds. Following the nomenclature [29] used, there is one variable to be considered in Eq (7), sub-vector $Z^T$, which indicates whether the bulls of each breed are proven. Due to the decrease in informational asymmetry and proof of the bull's genetic transmission capacity, it is expected that the semen dose from proven animals will have a higher price.

The second model aimed to explain the existence of a price differential for breeds and artificial insemination (AI) firms. Therefore, the specification of Eq (7) has two attributes: sub-vector $Z^B$ for breeds and sub-vector $Z^A$ for AI firms. According to [50], a product can be described according to its inherent attributes, thereby generating the need for a basic unit of analysis. For this analysis, the Gir breed and the AI firm ALTA were chosen as the baseline characteristics of the semen. Alternatively, the complete vector of characteristics could be considered in the estimation, excluding the constant. Information about this breed and the selected AI firm form the characteristic product (semen), and the difference in price obtained is relative to the price of semen from the Gir bull negotiated by the AI firm ALTA. This procedure also aims to avoid multicollinearity.

The third model was developed to identify the specificities of each breed. The attributes evaluated in this model were computed based on the progeny test. Considering the specification of the third model, one was estimated for each breed. As the test data for the Guzerá breed were not obtained, in the specification of Eq (7) for the Gir and Girolando breeds, the following characteristics were considered: $Z^R$, $Z^P$, $Z^S$, and $Z^L$. The sub-vector $Z^R$ represents the number of herds. It is expected that the greater the number of herds, the higher the price of the semen. The sub-vector $Z^P$ represents the PTAs: PTA–Milk (volume), PTA–Age at First Birth, PTA–Fat, and PTA–Protein. Given that the payment model adopted by dairy companies in the country favors production volume, it is expected that, across all breeds, the doses from the bulls with the best results for PTA–Milk (volume) will be more expensive. The sub-vector $Z^S$ indicates whether the semen is sexed or not. Sexed semen is expected to have a positive effect on semen price, since this attribute guarantees female offspring. Finally, the sub-vector $Z^L$

indicates whether the semen came from a bull that was alive. The effect of this attribute on semen price is ambiguous. If the market understands that the semen was from a bull that is not alive, this represents a shortage, and the effect on the price of semen tends to be positive; if the bull is alive, then the effect is negative. If the market understands that the semen from a (not) live bull is of better quality than semen from bulls from past generations, the effect of the bull not being alive (or being alive) on the price of the semen tends to be negative (positive).

The relationship between the price of the product and its attributes as established in Eq (6) has no theoretically defined functional form. Thus, the literature suggests the adoption of linear and semi-logarithmic functions and the Box-Cox transformation [49]. As an initial step, the literature suggests performing the Box-Cox test as a parameter for indicating the best functional form. The Box-Cox test performed in the present study did not indicate the functional form with more robust results. Thus, we followed the recommendation to use the semi-logarithmic specification of the hedonic price equation [28, 29, 33–35, 46, 49].

To estimate Models 1, 2, and 3, we have the following specifications, respectively:

$$ln(p_i) = \beta_0 + \beta_t Z^T + \varepsilon_i \tag{8}$$

$$ln(p_i) = \beta_0 + \sum_{b=1}^{B} \beta_b Z_b^B + \sum_{a=1}^{A} \beta_a Z_a^A + \varepsilon_i \tag{9}$$

$$ln(p_i) = \beta_0 + \beta_r Z^R + \sum_{p=1}^{P} \beta_p Z_p^P + \beta_s Z^S + \beta_l Z^L + \varepsilon_i \tag{10}$$

where $\beta$ represents the coefficients to be estimated, which are the implicit values associated with the different attributes, and $\varepsilon_i$ is the error term. In Eq (8) (Model 1), the sub-vector $Z^T$ indicates whether the progeny test was performed. In Eq (9) (Model 2), $Z^B$ is the breeds' sub-vector of attributes indexed by $b$, where $b = 1,\ldots, B$ (or $Z^B$ = *Girolando, Gir, Guzerá*), and $Z^A$ is the AI firm's sub-vector of attributes indexed by $a$, where $a = 1,\ldots, A$ (or $Z^A$ = *ALTA, CRV, ABS, SEMEX*). In Eq (10) (Model 3), the sub-vector $Z^R$ is the attribute of the number of herds, $Z^P$ is the sub-vector of attributes of PTAs indexed by $p$, where $a = 1,\ldots, P$ (or $Z^P$ = *PTA–Milk (volume), PTA–Age at First Birth, PTA–Fat, PTA–Protein*), $Z^S$ is the sub-vector of attributes that indicates whether the semen is sexed, and $Z^L$ is the sub-vector that indicates whether the semen is from a live bull.

The marginal value of each attribute is its implicit price, and it represents a change in the price given a marginal change in the characteristic [46]. Considering that the variable is continuous, the marginal value of this characteristic is the partial derivative of Eq (8), (9), and (10) in relation to that characteristic:

$$\frac{\partial ln(p)}{\partial z_i} = \left(\frac{\partial p}{\partial z_i}\right)\left(\frac{1}{p}\right) = \beta_i \tag{11}$$

In addition to calculating the implicit marginal price (Eq (11)), the marginal effect of each attribute on prices in percentage terms is calculated. Models that aim to explain the percentage impact on the dependent variable by means of dummy variables need to be corrected, as the variable is not directly related to a percentage. To accomplish this, the correction that [51] proposed, which is suitable for the semi-logarithmic form of adjustment of the variables in terms of elasticity, was used, as follows:

$$\hat{p} = 100 \times \left\{ \left[ exp\left(\hat{c} - \frac{1}{2}V(\hat{c})\right) \right] - 1 \right\} \tag{12}$$

where $\hat{p}$ is the marginal price of the attribute, $\hat{c}$ is the estimated coefficient, and $V(\hat{c})$ is the variance of $\hat{c}$.

Additionally, obtaining the values of the premiums or relative discounts of the attributes facilitates a comparison of the results of this study with those of other studies that may have been developed for other analysis contexts, such as different markets or other countries, breeds, or periods. More recent studies on the milk market have prioritized the relative effects of premiums on product attributes [28, 29, 48].

For the estimation of Eq (8), (9), and (10), we relied on ordinary least squares (OLS), considering all the characteristics described in Table 2. To identify the existence of heteroscedasticity in the estimated equations, the Breusch-Pagan/Cook-Weisberg test was performed. In cases where heteroscedasticity was found, White's robust correction process was applied.

## 3 Results

In 2017, 1.344 million doses of semen were collected in Brazil for the purpose of procuring milk, a number almost 6.5% higher than that observed in 2016. Another 2.957 million doses were imported for the same purpose. Approximately 6% of the total number of bulls used for milk production was via AI. Among the national breeds of milk bulls, Girolando stood out with 579,438 doses, Gir with 470,210 doses, and Guzerá with 58,379 doses [3]. The assessment carried out in this study precisely considers these breeds.

Initially, the implicit price of the semen of bulls that were proven (proven characteristic) was verified. This analysis strategy was carried out as an initial step and in an isolated way so that this characteristic could be evaluated, given that we did not obtain information about the PTA of Guzerá bulls (Table 3). Estimates were found considering all bulls (554 observations), Gir bulls (282), and Girolando bulls (231). The model also estimated for the Guzerá breed, however, due to the small sample size (available from the AI firms) of the proven bulls of this breed, the results did not turn out to be satisfactory. In the models for all bulls and for the Gir breed, robust White correction was required, as indicated by the Breusch-Pagan/Cook-Weisberg test. The proven characteristic was significant at the 1% level of significance in the three models and the estimated coefficients were positive, indicating a positive premium for the

**Table 3. Estimated parameters and implicit attribute prices for semen of dairy bulls.**

| Variable | All bulls | | | Gir breed | | | Girolando breed | | |
|---|---|---|---|---|---|---|---|---|---|
| | Coefficients (β) | Implicit price (R $/dose) | Relative Effect (%) | Coefficients (β) | Implicit price (R $/dose) | Relative Effect (%) | Coefficients (β) | Implicit price (R $/dose) | Relative Effect (%) |
| Proven | 0.393*** | 47.39 | 47.41 | 0.461*** | 66.59 | 57.29 | 0.295*** | 22.14 | 32.86 |
| | (0.095) | | | (0.125) | | | (0.146) | | |
| N | 554 | | | 282 | | | 231 | | |
| R² | 0.041 | | | 0.060 | | | 0.017 | | |
| F Statistic | 16.980 | | | 13.620 | | | 4.060 | | |
| Breusch-Pagan/Cook-Weisberg test for heteroscedasticity (χ2) | 24.950*** | | | 19.840*** | | | 0.920 | | |

Source: Own elaboration.

Dependent variable = lnPrice. Robust standard errors are in parentheses.

Significance level

*** 1%, ** 5%, * 10%.

Note: Implicit prices were calculated based on the average semen price for each characteristic.

semen of bulls with proven progeny. A premium for this attribute was estimated at +47.39 R $/dose (+47.41%) considering all bulls, +66.59 R$/dose (+57.29%) for the Gir breed and +22.14 R$/dose (+32.86) for the Girolando breed in relation to the price of semen from unproven bulls in each group (Table 3).

This estimate was expected because a gain in the amount charged was expected from the decrease in information asymmetry between the proven and unproven bulls. This finding corroborates what was observed by [23] about the commercial value of genetic evaluation in Gir bulls. This result also supports the hypothesis of [6], who found that genetic gain depends essentially on the genetic value of the bulls used, which makes it important to compare the genetic levels of the bulls from different herds as accurately as possible (Table 3).

The estimates of the second analysis sought to show the differentiation of prices from breeds and AI firms that sell bovine semen and are specialized in AI (Table 4). Baseline semen was defined as that from the Gir bull sold on AI firm ALTA. White's robust correction was performed, given the Breusch-Pagan/Cook-Weisberg test. The results of this analysis indicate that there is no significant premium on the price of semen of the Guzerá breed, but there is a significant and negative premium on the price of the Girolando breed (-6.82 R$/dose or -14.58%). Thus, the price of semen of the Guzerá breed is not statistically different from the semen of the Gir breed traded at ALTA, while the price of semen of the Girolando breed is statistically lower than the price of Gir semen traded at ALTA, ceteris paribus. There are positive and significant premiums from AI firms CRV (+58.86 R$/dose or +68.16%), ABS (+12.47 R $/dose or +30.45%) and SEMEX (+28.17 R$/dose or +54.90%) in relation to the price of semen of the Gir breed sold at ALTA, ceteris paribus (Table 4). This type of estimate is new in the literature because the work of [24] was carried out based on the data from only one AI firm (SEMEX) and one race (Holstein).

**Table 4. Estimated parameters and implicit attribute prices for semen of dairy bulls.**

| Breeds and AI firm's premium in relation to baseline semen (Gir bull from AI firm ALTA) | | | |
|---|---|---|---|
| Variable | Coefficients (β) | Implicit price (R$/dose) | Relative Effect (%) |
| Guzerá | -0.022 | -1.39 | -2.82 |
|  | (0.117) |  |  |
| Girolando | -0.156** | -6.82 | -14.58 |
|  | (0.068) |  |  |
| CRV | 0.524*** | 58.86 | 68.16 |
|  | (0.091) |  |  |
| ABS | 0.269*** | 12.47 | 30.45 |
|  | (0.073) |  |  |
| SEMEX | 0.441*** | 28.17 | 54.90 |
|  | (0.083) |  |  |
| N | 554 |  |  |
| $R^2$ | 0.081 |  |  |
| F Statistic | 10.120 |  |  |
| Breusch-Pagan/Cook-Weisberg test for heteroscedasticity (χ2) | 36.330*** |  |  |

Source: Own elaboration.

Dependent variable = lnPrice. Robust standard errors are in parentheses.

Significance level

*** 1%

** 5%, * 10%.

Note: Implicit prices were calculated based on the average semen price for each characteristic.

**Table 5. Estimated parameters and implicit attribute prices for semen of dairy bulls.**

| Variable | Gir breed | | | Girolando breed | | |
|---|---|---|---|---|---|---|
| | Coefficients (β) | Implicit price (R $/dose) | Relative Effect (%) | Coefficients (β) | Implicit price (R $/dose) | Relative Effect (%) |
| Number of Herds | 0.002*** | - | 0.19 | 0.009 | - | 0.97 |
| | (0.001) | | | (0.009) | | |
| PTA—Milk (volume) | 0.002*** | - | 0.15 | 0.001*** | - | 0.07 |
| | (0.000) | | | (0.000) | | |
| PTA—Age at First Birth | -0.005 | - | -0.47 | 0.004 | - | 0.40 |
| | (0.003) | | | (0.003) | | |
| PTA—Fat (kg) | 0.010 | - | 0.97 | - | - | - |
| | (0.013) | | | | | |
| PTA—Protein (kg) | -0.003 | - | -0.25 | - | - | - |
| | (0.019) | | | | | |
| Sexed | 1.693*** | 817.91 | 439.58 | 1.698*** | 506.96 | 434.49 |
| | (0.119) | | | (0.196) | | |
| Alive | -0.572*** | -27.99 | -44.25 | -0.263* | -9.97 | -23.82 |
| | (0.159) | | | (0.147) | | |
| N | 86 | | | 28 | | |
| R² | 0.826 | | | 0.8774 | | |
| F Statistic | 52.88 | | | 28.35 | | |
| Breusch-Pagan/Cook-Weisberg test for heteroscedasticity (χ2) | 2.88 | | | 8.5*** | | |

Source: Own elaboration.

Dependent variable = lnPrice. Robust standard errors are in parentheses.

Significance level

*** 1%, ** 5%

* 10%.

Note: Implicit prices were calculated based on the average semen price for each characteristic.

On the other hand, the last analysis sought to verify the relationship between the price of the semen doses of the bulls and the characteristics extracted from their genetic tests (Table 5). This analysis sought to complement those carried out previously, in order to present more concrete information on the pricing of semen according to the observed characteristics of the bulls. Normally, the dissemination of bulls for milk is carried out through catalogs that include various information of the animals, such as kinship, performance in milk production in previous years, prizes in dairy tournaments and more recently, genetic information obtained through the tests developed by Embrapa Gado de Leite.

In the last analysis, only bulls of the Gir and Girolando breeds that are already proven were considered in the sample. No implicit prices were obtained for each test, given that the valid information to generate the implicit price is the average price of the semen from the proven bull. The results for this characteristic, that is, semen from a proven bull was performed in the first analysis (Table 3). The estimates for both breeds sought to verify the effects of genetic tests on PTA—Milk, PTA—Age at First Birth, PTA—Fat (kg), PTA—Protein (kg) and Sexed on the price of semen, in addition to the effect of the bull being alive and the number of herds. For the Gir breed, there was a positive and significant effect on the price of PTA—Milk only from genetic tests, while the other tests were not significant. This result indicates that only PTA—Milk has an influence on the price of semen in the case of the Gir breed (+0.15%), ceteris paribus. The same result was obtained for the number of herds (+0.19%). There was also a positive

and significant effect of the semen being of the sexed type (+817.91 R\$/dose or + 439.58%). A negative and significant effect on the price of semen was obtained for the bulls that are alive (-27.99 R\$/dose or -44.25%). The estimation for the Girolando breed indicates a positive and significant effect only of PTA—Milk on the price of semen (+0.07%). The sexed semen coefficient was also positive and significant (+506.96 R\$/dose or + 434.49%), while the semen coefficient of alive bulls was negative and significant (-9.97 R\$/dose or -23.82%) (Table 5).

As for the PTA, these results indicate that only the PTA—Milk is significant in the formation of the semen price. These estimates are new in the literature and the result was expected for PTA—Milk, as the producers want to produce the largest volume of milk possible. The results for PTA–Fat and PTA–Protein indicate that perhaps the producers do not observe the amount of fat and protein in milk potentially produced from females born from semen acquired for AI. [24] found that volume, protein, and fat content were significant in the pricing of dairy bulls in their analysis.

Quality is of interest to the dairy industry because lower bacterial load increases the shelf life of milk. A few dairy companies in Brazil adequately remunerate milk with more solids [52]. The low bonus for protein and fat in milk implies paying these products at the same price as for water, lactose and minerals, and the nutritional requirements are much higher for the first two. The energy required to produce a kilo of fat is 56 times greater than that required for a kilo of water, lactose, and minerals, and a kilo of protein is 28 times greater. Thus, if there is no adequate bonus, it seems uneconomic for the producer to produce fat and protein, and it is preferable, on the contrary, to produce milk as watery as possible [53]. This is the justification for the rationality of not paying more for semen from bulls with good performances in genetic evaluations to produce solids. [54] indicated, using a Holstein bulls selection index for 15 countries, that the emphasis on milk production (volume) had a weight of almost 60%.

The result for the semen that guarantees the gestation of a female (sexed) was also expected. [55] had already indicated that genetic progress would be maximized in breeding programs for milk production in which the sexual proportion was controlled at the time of artificial insemination, obtaining males or females, as desired. The positive and significant result of the number of herds indicates that a greater confidence can be obtained from an increase in the number of generations of those bulls. The negative and significant effect of the price of semen from live bulls indicates a lesser restriction of semen supply from these bulls, in contrast to the semen of dead bulls that may have a reduced supply. [24] found a similar result. However, the expected result was that live bulls would have a positive premium on prices. Due to the continuous genetic improvement, bulls of more recent generations tend to genetically transmit more positive performance on the characteristics of milk such as lactation, milk production (volume) and solid content. This result is indicative of information asymmetry in the semen market of dairy bulls.

## 4 Discussion and implications

Techniques based on selection and genetic improvement are increasingly being used in the production process, especially in the livestock environment in Brazil. Genetic herd mapping is among the most commonly used techniques to increase milk production and productivity. Mapping makes it possible to identify the main genetic characteristics that can positively influence milk production. It is not only done with cows but also with semen-supplying bulls that can genetically transmit traits to female offspring, affecting their potential to produce milk. Large volumes of data are being collected on the possible attributes of bulls and their semen, such as the transmission of the ability to produce greater volumes of milk and sexed semen. However, the market still prices bulls' semen without reference to the intrinsic value of each

attribute linked to the semen. This paper contributes to this aspect by providing information about the implicit prices attributed to the characteristics of bulls and their semen. AI is being increasingly used by dairy producers seeking to increase production and obtain optimum production levels to remain competitive in the market; bovine semen is one of the main inputs in AI.

This paper provides essential information, especially for dairy producers that buy bovine semen, about the value of semen attributes. Specifically, the information presented here can help them plan how much semen to buy from which bulls and how much to pay for sexed semen that has the greatest potential to increase milk volume or milk solids content. One of this study's main findings is that only the testing of the bull has a positive effect on the price of semen. This was expected because testing is an important step in genetic improvement and contributes to reducing the asymmetry of information about bulls' characteristics. There is no similar finding in the literature. Other evidence emerging from this research is that semen prices can vary between breeds and AI firms. Each breed has a different vocation for milk production, and this can impact the price of semen. Moreover, milk producers who want to buy semen should be aware of the various companies that sell and perform insemination, as there may be significant price differences between them. This is also new evidence in the literature.

It is also worth noting that for the breeds analyzed in Brazil, sexed semen tended to have a higher price, given the ability to transmit the potential to produce higher volumes of milk. This result is similar to that obtained by [24], but it is worth remembering that this study was conducted in another country and involved a different breed. Nevertheless, the evidence presented in this paper indicates that there is a greater and more significant appreciation in the Brazilian market for the ability to generate higher volumes of milk compared to higher milk solids content. This is new evidence in the literature on Brazilian dairy bull semen prices that can inform dairy producers' semen purchase decision making. It is worth stressing, again, that given intensifying market competitiveness, genetic improvement techniques are essential for producers. According to an [56] report, in the second half of 2021, 62% of Brazilian municipalities used doses of semen from dairy bulls, and the number of doses sold grew by more than 30% in 2020. Therefore, all the evidence presented in this paper contributes to decision making in a multimillion-dollar market concerning an essential input for the milk production process in Brazil.

## 5 Conclusions

The conclusions of this paper suggest means by which the bovine semen market can incentivize milk producers, based on the study's findings. This study found evidence that one of the variables that significantly and positively contributes to the premium attributed to the semen are those in which the semen was from proven bulls. This emphasizes the importance of carrying out genetic tests that provide information about the transmission capacity of the bull. The decrease in informational asymmetry means that there is a greater willingness to pay for the semen dose from proven bulls. The Sexed variable is the one with the highest estimated premium. The only significant milk PTA variable was volume. The evidence found for these variables is related to the profitability of the producer (buyer of the doses). The bonus paid by dairy companies in Brazil is mainly for volume and not quality. If the producer wants to receive more for his milk, he must produce more and there are no incentives for the improvement of milk solids. Thus, the producer pays more for the doses of semen that supply animals generating more milk and value the females more.

If dairy companies are interested in higher quality milk, they need to change the incentives given to producers, with investment in the bonus for solids, or simply payment programs per kilo of solid and not per liter of milk. The acquisition of semen with better performance in the

solids content by milk producers could imply a reduction in costs by dairy companies in the production of powdered milk (increased economic efficiency). As powdered milk is used by other food companies as a production input, there could be a cheaper price for the entire production chain and the product for final consumption. The strategy of increasing the solids content per liter of milk is already a common practice in countries such as New Zealand.

## Supporting information

**S1 Data.**
(XLSX)

## Acknowledgments

The authors would like to thank Embrapa Gado de Leite, Conselho Nacional de Desenvolvimento Científico e Tecnológico (CNPq), Coordenação de Aperfeiçoamento de Pessoal de Nível Superior (CAPES), Fundação de Amparo à Pesquisa do Estado de Minas Gerais (FAPEMIG), and Universidade Federal de Juiz de Fora (UFJF).

## Author Contributions

**Conceptualization:** Glauco Rodrigues Carvalho, Weslem Rodrigues Faria, Vinícius Pimenta Delgado Ribeiro Nardy.

**Data curation:** Glauco Rodrigues Carvalho, Weslem Rodrigues Faria, Vinícius Pimenta Delgado Ribeiro Nardy, Admir Antonio Betarelli Junior.

**Formal analysis:** Glauco Rodrigues Carvalho, Weslem Rodrigues Faria, Vinícius Pimenta Delgado Ribeiro Nardy, Admir Antonio Betarelli Junior.

**Funding acquisition:** Glauco Rodrigues Carvalho.

**Investigation:** Glauco Rodrigues Carvalho, Weslem Rodrigues Faria, Vinícius Pimenta Delgado Ribeiro Nardy, Admir Antonio Betarelli Junior.

**Methodology:** Glauco Rodrigues Carvalho, Weslem Rodrigues Faria, Vinícius Pimenta Delgado Ribeiro Nardy, Admir Antonio Betarelli Junior.

**Resources:** Glauco Rodrigues Carvalho, Weslem Rodrigues Faria, Vinícius Pimenta Delgado Ribeiro Nardy, Admir Antonio Betarelli Junior.

**Software:** Glauco Rodrigues Carvalho, Weslem Rodrigues Faria, Vinícius Pimenta Delgado Ribeiro Nardy, Admir Antonio Betarelli Junior.

**Supervision:** Glauco Rodrigues Carvalho, Weslem Rodrigues Faria, Vinícius Pimenta Delgado Ribeiro Nardy, Admir Antonio Betarelli Junior.

**Validation:** Glauco Rodrigues Carvalho, Weslem Rodrigues Faria, Vinícius Pimenta Delgado Ribeiro Nardy, Admir Antonio Betarelli Junior.

**Visualization:** Glauco Rodrigues Carvalho, Weslem Rodrigues Faria, Vinícius Pimenta Delgado Ribeiro Nardy, Admir Antonio Betarelli Junior.

**Writing – original draft:** Glauco Rodrigues Carvalho, Weslem Rodrigues Faria, Vinícius Pimenta Delgado Ribeiro Nardy, Admir Antonio Betarelli Junior.

**Writing – review & editing:** Glauco Rodrigues Carvalho, Weslem Rodrigues Faria, Vinícius Pimenta Delgado Ribeiro Nardy, Admir Antonio Betarelli Junior.

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
