## [Editor Report · Decision Letter 0]

12 May 2021

PONE-D-21-07942

Hedonic pricing analysis for semen of dairy bulls in Brazil

PLOS ONE

Dear Dr. Faria,

Thank you for submitting your manuscript to PLOS ONE. After careful consideration, we feel that it has merit but does not fully meet PLOS ONE’s publication criteria as it currently stands. Therefore, we invite you to submit a revised version of the manuscript that addresses the points raised during the review process.

While I find the topic is interesting, most of the papers cited in the manuscript are too old. I suggest the authors add some recent publications in their reference related to the topic and hedonic price models before I send out the paper for review. 

I am looking forward to receiving your manuscript with updated references. 

We look forward to receiving your revised manuscript.

Kind regards,

Zhifeng Gao

Academic Editor

PLOS ONE

Journal Requirements:

2. Thank you for stating the following financial disclosure: 'No'
---

## [Author Response · Author response to Decision Letter 0]

17 Jun 2021

Dear Editor:

We thank you in advance for taking the time to review our manuscript.

We agree with the editor’s point. We revised the manuscript to cite papers recently published in the literature on the topic of this paper and hedonic pricing models. The main changes were made in the introduction of the paper between pages 3 and 7 of the revised version of the manuscript. This section has been heavily updated due to the modifications. A modification was also made to section 2 on page 10 of the revised version of the manuscript. The list of references was also updated between pages 21 and 25. We hope that this version is able to proceed in the evaluation process. We are at the editor’s disposal to make additional modifications if necessary.

---

## [Decision Letter · Decision Letter 1]

21 Dec 2021

PONE-D-21-07942R1Hedonic pricing analysis for semen of dairy bulls in BrazilPLOS ONE

Dear Dr. Faria,

Thank you for submitting your manuscript to PLOS ONE. After careful consideration, we feel that it has merit but does not fully meet PLOS ONE’s publication criteria as it currently stands. Therefore, we invite you to submit a revised version of the manuscript that addresses the points raised during the review process. Please carefully review both reviewers' comments in your next round submission. Please address these comments item by item and explain what you have done in your response letter. 

We look forward to receiving your revised manuscript.

Kind regards,

Zhifeng Gao

Academic Editor

PLOS ONE

Journal Requirements:

Reviewers' comments:

Reviewer's Responses to Questions

**Comments to the Author**

1. If the authors have adequately addressed your comments raised in a previous round of review and you feel that this manuscript is now acceptable for publication, you may indicate that here to bypass the “Comments to the Author” section, enter your conflict of interest statement in the “Confidential to Editor” section, and submit your "Accept" recommendation.

Reviewer #1: (No Response)

Reviewer #2: (No Response)

2. Is the manuscript technically sound, and do the data support the conclusions?

Reviewer #1: Yes

Reviewer #2: Yes

3. Has the statistical analysis been performed appropriately and rigorously? 

Reviewer #1: Yes

Reviewer #2: Yes

4. Have the authors made all data underlying the findings in their manuscript fully available?

Reviewer #1: Yes

Reviewer #2: Yes

5. Is the manuscript presented in an intelligible fashion and written in standard English?

Reviewer #1: Yes

Reviewer #2: Yes

6. Review Comments to the Author

Reviewer #1: This study uses a hedonic price model to analyze the price of semen doses of dairy bulls using the seller’s data from Brazil. The study infers the relationship between the characteristics of the bull and the price of the semen. It also quantifies how the attributes affect the price of semen from milk bulls, and identifies whether proven bulls have higher prices than the others. The authors managed to find there is an additional premium for proven bulls, found that semen doses from Gir bulls proved to be more expensive, and concluded the most valued feature is the dose being sexed. Despite the methodology used in this interesting topic, this article could be further improved in the following aspects.

2.2 model

I felt this part could be more specific in terms of the model specifications. The article would be improved if the authors made it clearer what econometric model finalized used in the manuscript in addition to the economic model and general introduction of the hedonic model. Readers would have to figure it out by reading the results, while the model part should be discussing it more clearly.

4. Conclusions

The ending of the manuscript is a bit too quick. As the ending part, the manuscript would benefit more from an extended discussion of how the current results compared with the previous ones, what is the significant contribution of the current study, and why it is so important. Those three components could become a Discussion & Implication section, in addition to the current final summary of the general results. Right now, it lacks the discussion of the broader impacts of the study, and readers may feel unsure why we should even care about this study.

Reviewer #2: Overall Paper Comments:

The authors use a hedonic pricing model to determine what value buyers of dairy bull semen place on certain characteristics of the semen given its genetic merit for milk production and quality. It was determined that milk volume is the pace setter for determining semen value. The research idea is not novel in and of itself. However, it does likely contribute to the literature from a regional aspect in that this information could be important for Brazilian dairy producers, sellers of semen, and milk purchasers. The findings of this research were not unexpected, but it provides the foundation on which further research can be based. Similarly, this foundation could lead to changes within the Brazilian dairy industry if milk components become more desirable by milk purchases than just volume of fluid milk. The reviewer realizes the authors were instructed to expand on the introduction and add some literature review better form the research objective. However, the introduction was a little long for my taste. The editor and the authors can discuss that topic if it is found to be worthy of discussion. Otherwise, the paper is acceptable. A few specific errors have been noted, and there may have been some overlooked during the review.

Specific Comments:

Page 3, Line 4

Should be “head” instead of “heads”.

Page 4, Line 7

Should be “possible to obtain instead of “possible obtain”.

Page 4, Line 21

Delete “that” following the word “but”.

Page 5, Line 14

A comma is needed prior to “because”.

Page 6, Line 15

Should be “the Holstein breed” instead of “Holstein breed”.

7. PLOS authors have the option to publish the peer review history of their article (what does this mean?). If published, this will include your full peer review and any attached files.

Reviewer #1: No

Reviewer #2: No

---

## [Author Response · Author response to Decision Letter 1]

17 Feb 2022

Dear Reviewer,

Thank you for your comments and suggestions. We would like to emphasize that all comments have been considered and that we are confident that all requests for modifications have been met. You will find a specific answer in red to each of the comments. In addition, each modification is highlighted in the revised version of the manuscript.

Regards,

Authors.

---

## [Decision Letter · Decision Letter 2]

4 Apr 2022

Hedonic pricing analysis for semen of dairy bulls in Brazil

PONE-D-21-07942R2

Dear Dr. Faria,

We’re pleased to inform you that your manuscript has been judged scientifically suitable for publication and will be formally accepted for publication once it meets all outstanding technical requirements.

Kind regards,

Zhifeng Gao

Academic Editor

PLOS ONE

Additional Editor Comments (optional):

Reviewers' comments:

Reviewer's Responses to Questions

**Comments to the Author**

1. If the authors have adequately addressed your comments raised in a previous round of review and you feel that this manuscript is now acceptable for publication, you may indicate that here to bypass the “Comments to the Author” section, enter your conflict of interest statement in the “Confidential to Editor” section, and submit your "Accept" recommendation.

Reviewer #1: All comments have been addressed

Reviewer #2: All comments have been addressed

2. Is the manuscript technically sound, and do the data support the conclusions?

Reviewer #1: Yes

Reviewer #2: Yes

3. Has the statistical analysis been performed appropriately and rigorously? 

Reviewer #1: Yes

Reviewer #2: Yes

4. Have the authors made all data underlying the findings in their manuscript fully available?

Reviewer #1: Yes

Reviewer #2: Yes

5. Is the manuscript presented in an intelligible fashion and written in standard English?

Reviewer #1: Yes

Reviewer #2: Yes

6. Review Comments to the Author

Reviewer #1: The authors addressed my last round of concerns by improving the model specification and conclusion part by adding a lot of materials and discussions. The revised manuscript is in a good shape and I would suggest accepting it for publication.

Reviewer #2: I appreciate the increase in detail as it relates to economic theory and statistical approach. I think one could follow it in the previous versions, but the detail is beneficial.

7. PLOS authors have the option to publish the peer review history of their article (what does this mean?). If published, this will include your full peer review and any attached files.

Reviewer #1: No

Reviewer #2: No

---

## [Editor Report · Acceptance letter]

11 Apr 2022

PONE-D-21-07942R2 

Hedonic pricing analysis for semen of dairy bulls in Brazil 

Dear Dr. Faria:

I'm pleased to inform you that your manuscript has been deemed suitable for publication in PLOS ONE. Congratulations! Your manuscript is now with our production department. 

Kind regards, 

on behalf of

Dr. Zhifeng Gao 

Academic Editor

PLOS ONE